# Equilibrium Aggregation: Encoding Sets via Optimization

Sergey Bartunov[1,2,*]        Fabian B. Fuchs[1,*]        Timothy P. Lillicrap[1]

[1]DeepMind, London, United Kingdom
[2]Now at CHARM Therapeutics, London, United Kingdom,

[*]Joint first authorship

## Abstract

Processing sets or other unordered, potentially variable-sized inputs in neural networks is usually handled by *aggregating* a number of input tensors into a single representation. While a number of aggregation methods already exist from simple sum pooling to multi-head attention, they are limited in their representational power both from theoretical and empirical perspectives. On the search of a principally more powerful aggregation strategy, we propose an optimization-based method called Equilibrium Aggregation. We show that many existing aggregation methods can be recovered as special cases of Equilibrium Aggregation and that it is provably more efficient in some important cases. Equilibrium Aggregation can be used as a drop-in replacement in many existing architectures and applications. We validate its efficiency on three different tasks: median estimation, class counting, and molecular property prediction. In all experiments, Equilibrium Aggregation achieves higher performance than the other aggregation techniques we test.

## 1  INTRODUCTION

Early neural networks research focused on processing fixed-dimensional vector inputs. Since then, advanced architectures have been developed for processing fixed-dimensional data efficiently and effectively. This format, however, is not natural for applications where inputs do not have a fixed dimensionality, are unordered, or have both of these properties. A strikingly successful strategy for tackling this issue has been to process such inputs with a series of aggregation $\rightarrow$ transformation operations.

An *aggregation* operation compresses a set of input tensors into a single representation of a known, predefined

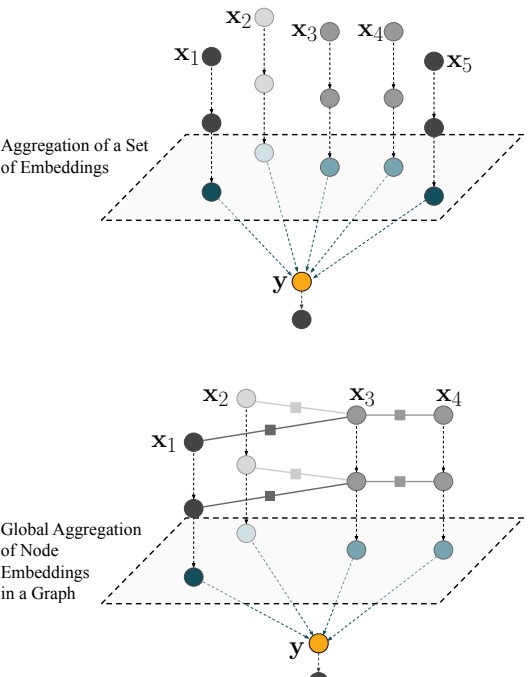

Figure 1: Global aggregation layers in typical neural networks for sets (top) and graphs (bottom). **Top**: each input set element $\mathbf{x}_i$ is first processed individually before being pooled into a global representation $\mathbf{y}$. This is followed by a final transformation block. **Bottom**: for graph data, the first part of the network is replaced by a graph or message passing neural network, but the global aggregation step is similar. In both cases, the global aggregation step drastically reduces the number of embeddings from many to one, rendering the right choice of aggregation technique critical for good model performance. The aggregation layer is typically implemented using sum-, max-, or attention-pooling. We propose a new aggregation mechanism, called Equilibrium Aggregation.

*Accepted for the 38th Conference on Uncertainty in Artificial Intelligence*  (UAI 2022).

dimensionality that can be then further sent to the downstream *transformation* block. Since the latter deals with fixed-dimensional inputs with a defined ordering, it can profit from the variety of techniques available for vector-to-vector computations.

This pattern can be seen in many architectures. For instance, Deep Sets [Zaheer et al., 2017] builds a representation of a set of objects by first transforming each object and then summing their embeddings. Similarly, Graph Neural Networks [Kipf and Welling, 2016, Battaglia et al., 2018] use a message-passing mechanism, which amounts to aggregating the set of input messages received by each node from its neighbours and then transforming the aggregate into a new message on the next layer (local aggregation). In many cases, several message passing layers are then followed by a global aggregation layer, where all node embeddings are aggregated into one global embedding vector describing the entire graph. Finally, Transformers [Vaswani et al., 2017] use self-attention, a mechanism that allows each object in the input set to interact with every other object and update its embedding by aggregating value embeddings from the rest of the set.

Mathematically, the aggregation $\phi(X) = \mathbf{y}$ compresses the input set $X = \{\mathbf{x}_1, \mathbf{x}_2, \ldots, \mathbf{x}_N\} \in 2^{\mathcal{X}}$ into a $D$-dimensional vector $\mathbf{y} \in \mathbb{R}^D$. In the case of Deep Sets [Zaheer et al., 2017] with sum aggregation, this reads

$$\phi(X) = \rho(\sum_{i=1}^{N} f(\mathbf{x}_i)), \tag{1}$$

where $f$ and $\rho$ are the optional input and output transformations, respectively.

Besides yielding a fixed-dimensional output embedding, (1) enforces an important inductive bias: permutation invariance. Global properties of sets or graphs (such as the free energy of a molecule) are independent of the ordering of the set elements. Taking advantage of such *task symmetries* [Mallat, 2016] can add robustness guarantees with respect to important classes of input transformations, and is known to help generalisation performance [Worrall et al., 2017, Weiler et al., 2018, Winkels and Cohen, 2018]. Other ways of incorporating permutation invariance are max-pooling, mean-pooling or attention aggregators [Kipf and Welling, 2016, Battaglia et al., 2018, Vaswani et al., 2017, Velickovic et al., 2018].[1]

However, it is exactly these aggregation functions which often introduce a bottleneck in the information flow [Zaheer et al., 2017, Wagstaff et al., 2019, Cai and Wang, 2020,

---

[1]Interestingly, even though in the case of Transformers for natural language processing the input is an ordered sequence, it appears beneficial to model the data as an order-independent set (or fully connected graph) with the sequential structure added via positional encodings.

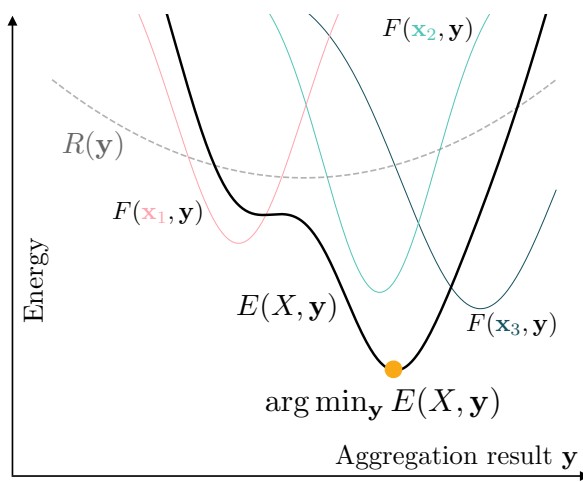

Figure 2: Schematic illustration of Equilibrium Aggregation. Each input $\mathbf{x} \in X$ contributes a potential value $F(\mathbf{x}, \mathbf{y})$ which are summed over the set $X$ and, together with the regularizer $R(\mathbf{y})$, form the total energy. Equilibrium Aggregation seeks to minimize this energy and the found minimum serves as the aggregation result.

Chen et al., 2020, Wagstaff et al., 2021]. It is easy to see that sum aggregation may struggle to selectively extract relevant information from individual inputs or subsets and while methods like multi-head attention (effectively amounting to weighted mean per each head) partially address this issue, we believe there is a fundamental need for more expressive aggregation mechanisms.

Motivated by this need, we develop a method called Equilibrium Aggregation which is a generalization over existing pooling-based aggregation methods and can be obtained as an implicit solution to optimization-based formulation of aggregation. We further investigate its theoretical properties and show that not only it is a universal approximator of set functions but that it is also provably more expressive than sum or max aggregation in some cases. Finally, we validate our insights empirically on a series of experiments where Equilibrium Aggregation demonstrates its practical effectiveness.

## 2 EQUILIBRIUM AGGREGATION

Our insight for developing better aggregation functions is grounded in the fact that the standard, pooling-based aggregation methods can be recovered as solutions to a certain optimization problem:

$$\phi(X) = \arg\min_{\mathbf{y}} \sum_{i=1}^{N} F(\mathbf{x}_i, \mathbf{y}), \tag{2}$$

where $F(\mathbf{x}, \mathbf{y})$ is a *potential* function.

For example, with $F(\mathbf{x}, \mathbf{y}) = (\mathbf{x} - \mathbf{y})^2$ (and assuming $\mathcal{X} =$

$\mathbb{R}$), one obtains the *mean aggregation* $\phi(X) = \frac{1}{N} \sum_{i=1}^{N} \mathbf{x}_i$, more examples can be found in Table 1. A natural question following this observation arises: can a more interesting aggregation strategy be induced by other choices of the potential function $F(\mathbf{x}, \mathbf{y})$?

We propose a method called Equilibrium Aggregation that addresses this question by letting the potential be a learnable neural network $F_\theta(\mathbf{x}, \mathbf{y})$ parameterized by $\theta$ which takes a set element $\mathbf{x}$ and the aggregation result $\mathbf{y} \in \mathcal{Y} = \mathbb{R}^M$ as an input and outputs a non-negative real scalar expressing the degree of "disagreement" between the two inputs. By also adding a regularization term, we obtain the energy-minimization equation for Equilibrium Aggregation:

$$\phi_\theta(X) = \arg\min_{\mathbf{y}} E_\theta(X, \mathbf{y}),$$

$$E_\theta(X, \mathbf{y}) = R_\theta(\mathbf{y}) + \sum_{i=1}^{N} F_\theta(\mathbf{x}_i, \mathbf{y}), \qquad (3)$$

where for the scope of the paper the regularizer is simply $R_\theta(\mathbf{y}) = \text{softplus}(\lambda) \cdot ||\mathbf{y}||_2^2$. A graphical illustration for this construction can be found on Figure 2.

Interestingly, this makes the result of the aggregation $\mathbf{y}$ be defined *implicitly* and generally not available as a closed-form expression. Instead, one can find $\mathbf{y}$ by numerically solving the optimization problem (3), e.g., by gradient descent:

$$\mathbf{y}^{(t+1)} = \mathbf{y}^t - \alpha \nabla_{\mathbf{y}} E_\theta(X, \mathbf{y}^{(t)}), \quad \phi_\theta(X) = \mathbf{y}^{(T)}. \quad (4)$$

Under certain conditions and with a large enough number of steps $T$, this procedure provides a sufficiently accurate solution that is itself well-defined and differentiable: either explicitly, through the unrolled gradient descent [Andrychowicz et al., 2016, Finn et al., 2017], or via the implicit function theorem applied to the optimality condition of (3) [Bai et al., 2019, Blondel et al., 2021]. This allows to learn parameters of the potential $\theta$ and also to train the whole model involving the aggregation end-to-end.

In general, it is not guaranteed that gradient-based optimization will converge to the global minimum of (3) when the potential is an arbitrarily structured neural network. However, with a large enough regularization weight $\lambda$, it is possible to enforce convexity at least in the subspace of $\mathcal{Y}$ [Rajeswaran et al., 2019b]. When the gradient descent is initialized from a learnable starting point or, as in our implementation, from the zero vector, it becomes sufficient to find just a stationary point as long as the next layer in the network makes use of the aggregation result. Relaxing the need for convergence to the global minimum together with the use of flexible neural networks allows to implement a potentially complex and expressive aggregation mechanism. In our implementation we employ explicit differentiation through gradient descent and find that the network generally learns convergent dynamics (4) automatically, even with a fairly small number of iterations such as $T = 10$.

| Aggregation | $\phi(X)$ | $F(\mathbf{x}, \mathbf{y})$ |
|---|---|---|
| Mean | $\frac{1}{N} \sum_{i=1}^{N} \mathbf{x}_i$ | $(\mathbf{x} - \mathbf{y})^2$ |
| Median | $\mathbf{x}_{[N/2]}$ | $|\mathbf{x} - \mathbf{y}|$ |
| Max | $\max\{\mathbf{x}_1, \ldots, \mathbf{x}_N\}$ | $\max(0, \mathbf{x} - \mathbf{y})$ |
| Sum | $\sum_{i=1}^{N} \mathbf{x}_i$ or $\arg\min_{\mathbf{y}} \left[ \frac{\mathbf{y}^2}{2} + \Sigma_i F(\mathbf{x}_i, \mathbf{y}) \right]$ | $-\mathbf{x} \cdot \mathbf{y}$ |
| **Equilibrium Aggregation** | $\arg\min_{\mathbf{y}} E_\theta(X, \mathbf{y})$ | Neural network $F_\theta(\mathbf{x}, \mathbf{y})$ |

Table 1: A comparison between Equilibrium Aggregation and pooling-based aggregation methods. Equations are given for the scalar case or can be applied coordinate-wise in higher dimensions.

To additionally encourage convergence, we consider the following *auxiliary loss* that penalizes the norm of the energy gradient at each step of optimization:

$$L_{\text{aux}}(X, \mathbf{y}, \theta) = \frac{1}{T} \sum_{t=1}^{T} ||\nabla_{\mathbf{y}} E_\theta(X, \mathbf{y}^{(t)})||_2^2. \quad (5)$$

We simply add the auxiliary loss to the main loss incurred by the task of interest and optimize the sum during the training. We further empirically assess convergence of the inner-loop optimization in Section 5.3.

## 3 UNIVERSAL FUNCTION APPROXIMATION ON SETS

According to the universal function approximation theorem for neural networks [Hornik et al., 1989, Cybenko, 1989, Funahashi, 1989], an infinitely large multi-layer perceptron can approximate any continuous function on compact domains in $\mathbb{R}$ with arbitrary accuracy. In machine learning, we typically do not know the function we aim to approximate. Hence, knowing that neural networks can in theory approximate anything is comforting. Equally, we seek to build inductive biases into the networks in order to facilitate learning, using more sophisticated architectures than multi-layered perceptrons. It is imperative to be aware whether and to what extent those modifications restrict the space of learnable functions.

Similar constructions to Equilibrium Aggregation, i.e. optimization-defined models defined as $\mathbf{y} = \arg\min_{\mathbf{y}} G(X, \mathbf{y})$, have previously been studied in the literature, especially in the context of permutation-sensitive (i.e. *not* permutation invariant) functions [Pineda, 1987, Finn and Levine, 2017, Bai et al., 2019] and various results with respect to universal function approximation were obtained. It is not obvious, however, how these results translate to the important permutation-invariant case we consider in this paper. Introducing permutation invariance self-evidently restricts the space of functions that can be approximated. In the next section we directly

address the question of what set functions can be learned by Equilibrium Aggregations and establish a universality guarantee.

## 3.1 UNIVERSALITY OF EQUILIBRIUM AGGREGATION

In this section, we will see that Equilibrium Aggregation is indeed able to approximate all continuous permutation invariant functions $\psi$. We start by stating a few assumptions: We assume a fixed input set size $N$ of scalar inputs[2] $x_i$ (note the dropping of the boldface to indicate that these are not vectors anymore) and a scalar output. We further assume that input space $\mathcal{X}$ is a compact subset of $\mathbb{R}^N$. For simplicity, without loss of generality (as we can always rescale the inputs), we choose this to be $[0,1]^N$:

$$\psi : [0,1]^N \to \mathbb{R}. \qquad (6)$$

As $\psi$ is permutation invariant, the vector valued inputs can be seen as (multi)sets. For a discussion on why considering uncountable domains (i.e. the real numbers) is important for continuity, see Section 3 of Wagstaff et al. [2019].

We consider a neural network architecture with Equilibrium Aggregation as a global pooling operation of the following form:

$$\phi(X) = \rho(\arg\min_{\mathbf{y}} \sum_i F_\theta(x_i, \mathbf{y})), \qquad (7)$$

where $F_\theta$ (the potential function) and $\rho$ are modeled by neural networks, which are assumed to be universal function approximators. Note that, for simplicity of the proof, we implicitly set the regulariser to 0. We refer to the output of $\arg\min \sum_i F_\theta(x_i, \mathbf{y})$ as the *latent space*, analogous to the terminology used in Wagstaff et al. [2019] with respect to the Deep Sets architecture [Zaheer et al., 2017]. We prove the following:

**Theorem 1** *Let the latent space be of size $M = N$, i.e. $\mathbf{y} \in \mathbb{R}^N$. Then all permutation invariant continuous functions $\psi$ can be approximated with Equilibrium Aggregation as defined in* (7).

**Proof** For the purpose of this proof, we assume $F_\theta$ takes the form:

$$F_\theta(x_i, \mathbf{y}) = \sum_{k=1}^{M} (\frac{y_k}{N} - x_i^k)^2, \qquad (8)$$

where $k$ serves both as an index for the vector $\mathbf{y}$ and as an exponent for $x_i$. There are two sums now, an inner one in the

definition of $F_\theta$ and an outer one over the nodes in (7). Note that $F_\theta$ is continuous and can therefore be approximated by a neural network. Importantly, $F_\theta$ is also convex and can therefore assumed to be optimised with gradient descent to find $\arg\min(\mathbf{y})$. Note that all $M$ terms can be optimised independently as $X$ is fixed. It is a well-known fact that minimising the sum of squares yields the mean:

$$\arg\min_z \sum_{i=1}^{N} (z - x_i)^2 = \frac{1}{N} \sum_{i=1}^{N} x_i. \qquad (9)$$

It follows that minimising the sum of energies defined in (8) yields

$$y_k^{min} = \sum_i x_i^k \quad \text{for } k \in \{1, \dots, N\}. \qquad (10)$$

For inputs $(x_1, ..., x_M) \in [0,1]^M$, this mapping to $\mathbf{y}$ is evidently continuous and surjective with respect to its range $[0, M]^N$. We also know from Lemma 4 in Zaheer et al. [2017] that this mapping is injective and from Lemma 6 that it has a continuous inverse.[3] $\psi$ is continuous by definition and, therefore,

$$\rho = \psi \circ \left( \arg\min_{\mathbf{y}} \sum_i F_\theta(x_i, \mathbf{y}) \right)^{-1} \qquad (11)$$

is continuous[4] as long as the inputs $x_i$ are constrained to $[0,1]$ and can therefore be approximated by a neural network. However, via a global re-scaling of the inputs, this proof can be used for any bounded input domain. Hence, any permutation invariant, continuous $\psi$ on a bounded domain can be appoximated via Equilibrium Aggregation for a latent space of size $M = N$. ∎

## 3.2 COMPARISON TO DEEP SETS

So far, we have only been able to prove that Equilibrium Aggregation scales at least as well as Deep Sets. By that, we mean that universal function approximation can be achieved with $N = M$, i.e. having as many latents as inputs is *sufficient*. (For Deep Sets, we also know that $N = M$ is necessary [Wagstaff et al., 2019].) Even though we currently do not know whether it is possible to achieve universal function approximation with a smaller latent space, there is some indication that Equilibrium Aggregation may have more representational power, as we will lay out in the following:

Using one latent dimension, Deep Sets with max-pooling can obviously represent $\psi(X) = \max(X)$, but it cannot

---

[2]This is a common simplification in the literature on universal function approximation on sets. For a discussion on how to generalise from the scalar to the vector case, see Hutter [2020].

[3]We refer to Appendix B.4 in Wagstaff et al. [2019] as to why the term $k = 0$ in (10) can be dropped for fixed set sizes.

[4]The superscript $-1$ indicates the functional inverse w.r.t. $X$.

represent (or even approximate) the sum for set sizes larger than 1. Vice versa, sum-pooling can represent $\psi(X) = \text{sum}(X)$, but it cannot represent $\max(X)$ [Wagstaff et al., 2019]. Equilibrium Aggregation can represent both sum and max pooling, each with just one latent dimension (i.e. $\mathbf{y} \in \mathbb{R}^1$) as shown in Table 1.

# 4 RELATED WORK

Equilibrium Aggregations sits at the intersection of two machine learning research areas: aggregation functions and implicit layers. In the following, we give an overview over the work closest related in each of the fields, respectively.

## 4.1 AGGREGATION FUNCTIONS

Perhaps the most popular approach for obtaining a permutation invariant encoding of sets is Sum pooling. A particular instance of this is Deep Sets [Zaheer et al., 2017], as described in (1). A central finding of Wagstaff et al. [2019] is that the latent space, i.e. the dimensionality of the result of $\sum_i f(x_i) \in \mathbb{R}^M$ needs to be at least as large as the number of inputs $N$, i.e. $M \geq N$ in order to guarantee universal function approximation. This applies to many other aggregation methods as well and, to the best of our knowledge, there is currently no known pooling operation which does not introduce this scaling issue.

Principal Neighbourhood Aggregation (PNA) [Corso et al., 2020] addresses the limitations of each individual pooling operator such as Sum or Max by combining four different pooling operators and three different scaling strategies resulting into a simultaneous 12-way aggregation. Despite the more sophisticated aggregation procedure, Corso et al. [2020] come to very similar conclusions as Zaheer et al. [2017] and Wagstaff et al. [2019], namely that $N = M$ is both necessary and sufficient. They prove the necessity for any set of aggregators as well as the sufficiency for a specific set. In our work, we further expand this line of thinking by allowing the model to *learn* the desired aggregation operator which may include PNA or something drastically different.

Learnable Aggregation Functions (LAF) [Pellegrini et al., 2020] provide a similar framework for learning an aggregation operator by expressing it as a combination of several weighted $L_p$ norms, where the weights and the $p$ parameters are trained jointly with the model. Even though LAFs are capable of expressing operators used in PNA and beyond, it is not clear how they can reproduce other aggregation methods such as attention. In contrast, our method can learn attention (see Supplementary Material for details) as well as even more expressive aggregation functions.

Further generalization of the functional form of the aggregation operator leads to the Karcher or Fréchet mean [Grove

and Karcher, 1973], which are defined as a solution to the distance-generalization problem over a metric space $\mathcal{X}$ with a metric $d(\cdot, \cdot)$:

$$\bar{\mathbf{x}} = \arg \min_{\bar{\mathbf{x}} \in \mathcal{X}} \sum_{i=1}^{N} d^2(\bar{\mathbf{x}}, \mathbf{x}_i), \quad \mathbf{x}_i \in \mathcal{X}.$$

While closely related to the Karcher or Fréchet mean, Equilibrium Aggregation differs in not restricting the aggregation result to the same space as $\mathcal{X}$ and allowing radically non-symmetrical potential functions, together with the regularizer.

Finally, Janossy Pooling [Murphy et al., 2019] generalizes the idea of standard, coordinate-wise pooling to make use of higher-order interactions between set elements. Despite the potential for practical effectiveness, it is unclear whether these developments guarantee better approximation results in the general case [Wagstaff et al., 2021]. While Equilibrium Aggregation is also fully compatible with Janossy Pooling and may profit from even more expressive energy functions with pairwise or triplet interactions, this may not be necessary as such interactions can be emulated within the optimization process and ultimately come at a significant computational cost for larger set sizes.

In addition to formulating more expressive pooling operators, there is also a body of work concerned with multi-step parametric models for set encoding [Vinyals et al., 2015, Lee et al., 2019]. Inevitably, to achieve permutation invariance these models rely on some kind of a pooling as a building block, such as the ones outlined above. Equilibrium Aggregation being a drop-in replacement for sum- or attention-pooling can be used in those models, too.

## 4.2 IMPLICIT AND OPTIMIZATION-BASED MODELS

Gradient-based optimization has been utilized in a large number of applications [Amos, 2019]: image denoising [Putzky and Welling, 2017], molecule generation [Duvenaud et al., 2015, AlQuraishi, 2019], planning [Amos et al., 2018] and combinatorial search [Hottung et al., 2020, Bartunov et al., 2020] to name a few. While there is a large body of work where gradient descent dynamics is decoupled from learning, (e.g., Du and Mordatch [2019], Song and Ermon [2019], our work is particularly closely related to methods that seek to learn the underlying objective function end-to-end, such as Putzky and Welling [2017], Rubanova et al. [2021].

A closely-related family of methods involve the idea of defining computations inside a model *implicitly*, i.e. via a set of conditions that a particular variable must obey instead of prescribing directly how the variable's value should be computed. Deep Equilibrium Models (DEQs) formulate this via a fixed point of an update rule speci-

fied by the model [Pineda, 1987, Liao et al., 2018, Bai et al., 2019] and Implicit Graph Neural Networks explore this idea in the context of graphs [Gu et al., 2020]. Neural ODEs [Chen et al., 2018] allow to parametrize a derivative of a continuous-time function specifying the computation of interest. iMAML [Rajeswaran et al., 2019a] considers an implicit optimization procedure for the purpose of finding model parameters suitable for gradient-based meta-learning [Finn et al., 2017].

Our work is similar in spirit but focuses specifically on the aggregation block for encoding sets, which can be seen as a small but generic building block that can be combined with arbitrary model architectures. Similarly to OptNet [Amos and Kolter, 2017], we propose a layer architecture that can be used inside another implicit or traditional multi-layer neural network.

### 4.3 LEARNING ON DISTRIBUTIONS

An important use-case for set encoding is machine learning models aiming at learning a distribution from a finite sample. A recent example is Neural Processes [Garnelo et al., 2018], which builds a simple permutation-invariant representation of the training set via averaging of its encoded elements and a similar construction of Edwards and Storkey [2016]. Equilibrium Aggregation can be applied to building a more advanced variation on this idea that substitutes the average pooling with a maximum a posteriori (MAP) parameter estimation (see the Supplementary Material for details). It is also straight forward to replace the MAP formulation with the parametric variational inference approach, further bridging the gap between set encoding and distribution learning.

## 5 EXPERIMENTS

In this section, we describe three experiments with the goal of analyzing the performance of Equilibrium Aggregation in different tasks and comparing it to existing aggregation methods. Our intention is not to achieve state of the art results on any particular task. Instead, we strive to consider archetypal scenarios and applications in which performance significantly depends on the choice of aggregation method so it can be studied in isolation from other issues.

In all experiments we let the models to train for $10^7$ steps of Adam optimizer [Kingma and Ba, 2014]. Since maximizing performance is not the goal of our experiments, we do not perform an extensive hyperparameter search, only limiting it to a sweep over the learning rate (chosen from $\{10^{-4}, 3 \times 10^{-4}, 10^{-3}\}$) and the auxiliary loss weight (on MOLPCBA only). To that end, we use a small subset of the training set reserved for validation (Omniglot and MOLPCBA benchmarks only). We rely on a single GPU training regime using Nvidia P100s and V100s. All exper-

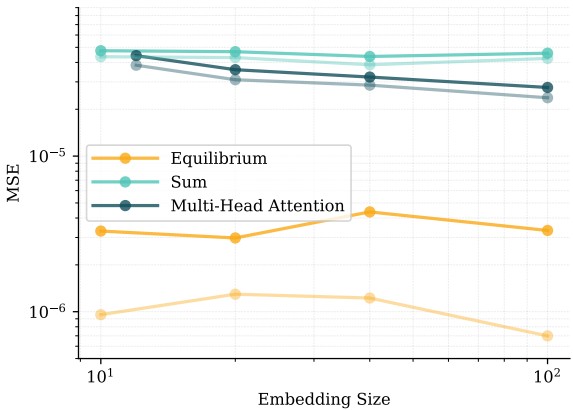

Figure 3: Median estimation of a 100-number set with three different aggregation methods. The bold lines correspond to the average performance over 5 seeds, the faded lines show the best performing seed of the respective model. Mean square error is computed for varied set embedding sizes on $8 \times 10^5$ number of sets.

imental code is written in Jax primitives [Bradbury et al., 2018] using Haiku [Hennigan et al., 2020]. Source code for the most crucial parts of our implementation can be found in the Supplementary Material.

### 5.1 MEDIAN ESTIMATION

In this experiment, the neural network is tasked with predicting the median value of a set of 100 randomly sampled numbers. Each set is sampled from either a Uniform, Gamma or Normal distribution with fixed parameters, similarly to Wagstaff et al. [2019]. The basic architecture for pooling-based aggregation baselines consists of first embedding each number in the set with a fully connected ResNet [He et al., 2016] with layer sizes $[256, 256, D]$, where $D$ is the set embedding size. Then, the embeddings are pooled with the corresponding method into a $D$-dimensional vector and the median is predicted from it using another fully connected network with layer sizes $[D, 128, 1]$. A simple square loss is used to regress the median.

Equilibrium aggregation, in contrast, performs the input encoding and aggregation simultaneously by doing a 5-step gradient optimization of (3) with the potential function implemented as a ResNet with layer sizes $[256, 256, 1]$ taking a $D + 1$-dimensional input ($D$ for the implicit aggregation result and 1 for the input number). The result is then also transformed into the prediction using the same output network as in the baseline methods.

We compare three models, Sum aggregation analogous to Deep Sets [Zaheer et al., 2017], Multi-head attention with 4 heads, each operating with $D/4$ dimensional keys, values and learned query vectors, and Equilibrium Aggregation

**Input**: a set of 16 images

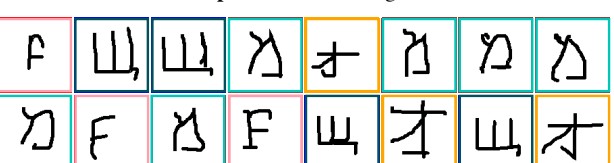

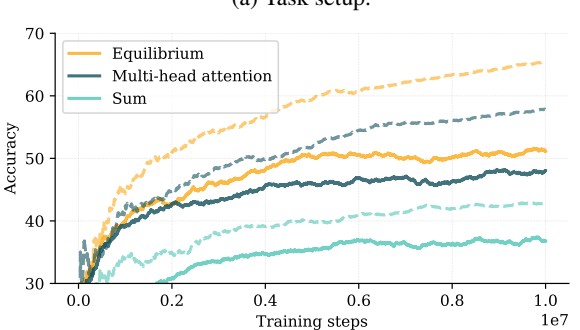

**Answer**: number of unique characters (**4** / 10)

(a) Task setup.

(b) Train (dashed) and test (solid) accuracy for different aggregation methods.

Figure 4: Omniglot class counting task.

as described above. For each of the models we vary the embedding size and assess the mean square error after $10^7$ training steps. Empirical results are shown on Figure 3.

Equilibrium aggregation achieves one (for average across 5 seeds) or two (for the best out of 5 seeds) orders of magnitude better estimation error than the baseline pooling methods which confirms its higher representational power in this simple setting. Importantly, in this experiment, there is no distinction between training and test distributions as the samples are continuously drawn and never repeated. Hence, we are primarily testing the representation power of the approaches as opposed to data efficiency in this particular example. However, it is worth noting that all architectures have roughly the same amount of trainable parameters. Presumably, the low error achieved by Equilibrium Aggregation suggests that it managed to discover or reasonably well approximate the analytical solution $F(\mathbf{x}, \mathbf{y}) = |\mathbf{x} - \mathbf{y}|$.

## 5.2 OMNIGLOT CLASS COUNTING

We proceed to the more challenging task of counting the number of unique character classes in a set of 16 Omniglot images, which is inspired by Lee et al. [2019]. Omniglot [Lake et al., 2015] is a dataset of handwritten characters that are organized into alphabets and then into character classes for each of which only 20 instances are available. We randomly choose between 1 and 10 character classes and sample their images to form the input set. The model then needs to aggregate those images and infer the num-

ber of unique character classes by outputting a vector of probabilities for each of the $1, 2, \ldots, 10$ possible number of classes (see Figure 4a for a visual illustration).

Original images are downsized to $32 \times 32$ and encoded using a convolutional ResNet with $[16, 32, 64]$ hidden channels in each of the three blocks correspondingly. Each block operates with $3 \times 3$ filters and a stride of 2 and hence reduces spatial sizes of the input tensor by half. The ResNet output is then flattened and linearly projected into a 256-dimensional input embedding. After the encoding step, as in the previous experiment, Sum, Multi-Head Attention with 4 heads and Equilibrium Aggregation perform set aggregation into 256-dimensional set embedding and predicted the number of classes using a simple softmax distribution using a fully-connected ResNet with layer sizes of $[128, 10]$. Equilibrium Aggregation also uses a ResNet potential with $[512, 512, 32]$ structure where the output of the last layer is squared and then summed to form a scalar potential value. We used 10 iterations of inner-loop optimization in this experiment.

Each model is trained on the characters from Omniglot train set for $10^7$ steps and with a batch size of 8. Train and test accuracies are reported in Figure 4b. One can see that, again, Equilibrium Aggregation outperforms both of the baselines, both in terms of train and test set accuracy. This shows that, on the one hand, Equilibrium Aggregation has a significantly larger capacity and thus better fits the training data. On the other hand, this capacity results into better generalization and, presumably, a more robust aggregation strategy.

## 5.3 GLOBAL AGGREGATION IN GRAPH NEURAL NETWORKS

Finally, we study the effect of different aggregation methods in the global *readout* layer of a graph neural network (GNN) on a well-established MOLPCBA benchmark [Hu et al., 2020]. In this task, the model is required to predict 128 global binary properties of an input molecule. This is traditionally implemented within the GNN framework by first applying several layers of message-passing on a graph and then aggregating the resulting 300-dimensional node embeddings into a single 300-dimensional graph representation from which the predictions are made. Since there is more than one prediction task per molecule, mean average precision (MAP) is used as an evaluation metric. The test MAP is reported for the best MAP attained on the validation set as the model is training. The validation and test metrics are periodically evaluated from model snapshots taken approximately every $10^4$ training steps.

For this experiment, we choose two popular GNN architectures, namely a Graph Convolutional Network (GCN) [Kipf and Welling, 2016] and a Graph Isomorphism Network (GIN) [Xu et al., 2018] that both use a simple Sum readout in their canonical implementations by Hu et al. [2020]. We

Table 2: Comparison between different aggregation methods on MOLPCBA.

| Local Aggregation | Global Aggregation | Validation MAP | Test MAP |
|---|---|---|---|
| Graph Convolutional Network [Kipf and Welling, 2016] | Sum | 0.223 | 0.203 |
| | Multi-Head Attention | 0.248 | 0.229 |
| | Principal Neighbourhood Aggregation | 0.226 | 0.209 |
| | **Equilibrium Aggregation** | **0.269** | **0.252** |
| Graph Isomorphism Network [Xu et al., 2018] | Sum | 0.255 | 0.232 |
| | Multi-Head Attention | 0.254 | 0.234 |
| | Principal Neighbourhood Aggregation | 0.262 | 0.244 |
| | **Equilibrium Aggregation** | **0.263** | **0.246** |
| **Equilibrium Aggregation** | **Equilibrium Aggregation** | **0.269** | **0.258** |

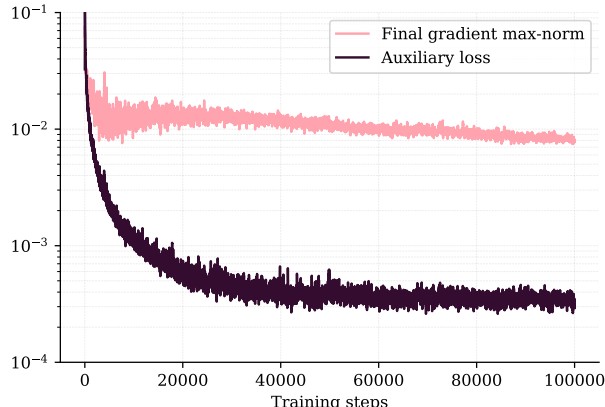

Figure 5: Inner-loop optimization statistics on MOLPCBA with the GIN architecture. The pink curve shows the maximum value of the $L^1$ norm along any dimension of the gradient on the last (15th) iteration of the inner loop. A value of $10^{-2}$ indicates a small gradient update and therefore good convergence of the optimizer. The dark purple curve tracks the auxiliary loss, i.e. the $L^2$ norm of the gradient update averaged across all 15 optimization steps (see (5)). Overall, these curves indicate stable, convergent behaviour despite a modest number of inner-loop optimization steps.

leave the architectures unchanged and only vary the global readout operation. Our implementation uses the Jraph library [Godwin* et al., 2020] and dynamic batch training with up to 8 graphs and 1024 nodes in a batch.

For the potential network we use an architecture similar to the previous experiment with layer sizes $[600, 300, 32]$, sum-of-the-squares output and 15 iterations for energy minimization.

The results are provided in Table 2. Overall, the empirical findings on MOLPCBA are consistent with the previous experiments with Multi-Head Attention providing a noticeable performance improvement over the basic Sum aggregation and Equilibrium Aggregation performing even better. In addition, we also evaluate Principal Neighbourhood Aggregation (PNA) [Corso et al., 2020], which has been proposed to address limitations an each individual pooling method in the context of GNNs and combines 12 combinations of scaled pooling methods. When combinining PNA with the GCN model, our experiments only show minor performance improvements over Sum pooling, in part because of increased overfitting. However, when applied to the GIN architecture, it achieves performance levels almost on par with Equilibrium Aggregation.

These results confirm one of the central hypotheses of this research: namely that the global aggregation of node embeddings is a critical step in graph neural networks. Perhaps surprisingly, the GCN generally benefited more from more advanced aggregation methods which is probably due to smaller number of parameters and thus decreased risk of overfitting. It is also worth noting that top performing GNN architectures achieve significantly higher test MAP on this task (see, e.g., Yuan et al. [2020], Brossard et al. [2020]).

In addition, we test an architecture where both local (i.e. node-level) and the global aggregations are performed using Equilibrium Aggregation. This model yields even better performance, albeit only marginally. While more careful architecture design that takes into account the specifics of Equilibrium Aggregation could potentially lead to larger per-

formance improvements, it should be noted that the molecular graphs in this task are relatively small and aggregation on the local level may be not the most critical step for a typical GNN.

Besides the task performance we also investigate the behaviour of the inner-loop optimization. Figure 5 plots two major statistics that quantify this: the max-norm of the final iterate of the optimization $\max_d |\nabla_{y_d} E(X, \mathbf{y}^{(T)})|$ and $L_{\text{aux}}$ (5). One can see that both rapidly decrease during the training and that a good degree of convergence is achieved. We observe similar behaviour with GCN and on other tasks we considered earlier.

# 6 DISCUSSION AND CONCLUSION

This work provides a novel optimization-based perspective on the widely encountered problem of aggregating sets that is provably universal. Our proposed algorithm, Equilibrium Aggregation, allows learning a problem-specific aggregation mechanism which, as we show, is beneficial across different applications and neural network architectures. The consistent empirical improvement brought by the use of Equilibrium Aggregation not only shows that many existing models are struggling from aggressive compression and inefficient representation of sets but also suggests a whole new class of set- or graph-oriented architectures that employ a composition of Equilibrium Aggregation operations. Beyond GNNs, other classes of models, such as Transformers, may also profit from more expressive aggregation operations, specifically in modelling long-term memory – a topic strongly connected to compression of sets [Rae et al., 2019, Bartunov et al., 2019], as well as potentially reduce the number of layers needed.

While there is a strong indication that using Equilibrium Aggregation as a building block is effective, the incurred computational cost may require more developments in differentiable optimization [Ernoult et al., 2020], architecture [Amos et al., 2017] and hardware design [Kendall et al., 2020], especially in order to compete with modern extra large models.

### Acknowledgements

We thank Peter Battaglia, Petar Veličković, Marcus Hutter, Yulia Rubanova and Marta Garnelo for their help with preparing the paper, insightful discussions and overall support during the course of the work.

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
