# OpenReview forum: "Equilibrium Aggregation: Encoding Sets via Optimization"
_auai.org/UAI/2022/Conference — UAI 2022 Poster_

### Official Review · Reviewer_AUC7 · 2022-04-03

**Q2(1) Originality/Novelty:** 3
**Q2(2) Significance/Impact:** 2
**Q2(3) Correctness/Technical Quality:** 3
**Q2(6) Clarity Of Writing:** 4
**Q6 Overall Score:** 6
**Q8 Confidence In Your Score:** 3

**Q1 Summary And Contributions:**

This paper proposes a novel aggregation method for processing sets and other unordered inputs (which require permutation invariance) in neural networks. The proposed approach is based on an optimization objective over a sum of potential functions of different inputs.

**Q2 Assessment Of The Paper:**

More detailed information regarding each of these aspects is given below:

**Q2(4) Quality Of Experiments (Optional):**

2: Fair: The experimental evaluation is weak: important baselines are missing, or the results do not adequately support the main claims.

**Q2(5) Reproducibility:**

4: Excellent: Key resources (e.g., proofs, code, data) are available and key details (e.g., proof sketches, experimental setup) are comprehensively described for competent researchers to confidently and easily reproduce the main results.

**Q3 Main Strengths:**

The main advantages of the proposed method are as follows:
1. The function aggregation problem, addressed in this paper, is an important one in the literature.
2. It is general; it can cover different aggregation strategies as special cases. This is because the potential functions in the objective are allowed to be learning via neural nets.
3. It is more efficient than the competing methods in some cases, with proven guarantees for the case M=N.
4. The dimension of the resulting function can be smaller than the dimension of inputs, which is an improvement over the other approaches from the literature which require this dimensionality to be at least as large as that of the inputs.

**Q4 Main Weakness:**

Numerical experiments are somewhat underwhelming since the hyper-parameters are not optimized/tuned; it's hard to judge if and by how much the proposed method outperforms the existing approaches from the literature. Does this also apply to competing methods, i.e. did the authors tune those methods or not?

Can you elaborate on this sentence: "It is also worth noting that top performing GNN architectures achieve significantly higher test MAP on this task".

**Q5 Detailed Comments To The Authors:**

Some minor typos/errors:
1. missing reference in page 6: "We compare three models, Sum aggregation inspired by [?], ..."
2. In page 5: "more sophisitaced aggregation procedure"
3. In page 2: "but that it also provably more expressive than sum or max aggregation in some cases."
4. In page 7: " one the one hand"
5. In page 8: "GNN achitectures achieve significantly higher"


**Q7 Justification For Your Score:**

This is an important problem in the literature and overall, the proposed approach seems general and applicable to different aggregation methods.

**Q9 Complying With Reviewing Instructions:**

1: Yes.

---

### Official Review · Reviewer_BFuE · 2022-04-20

**Q2(1) Originality/Novelty:** 2
**Q2(2) Significance/Impact:** 2
**Q2(3) Correctness/Technical Quality:** 4
**Q2(6) Clarity Of Writing:** 4
**Q6 Overall Score:** 5
**Q8 Confidence In Your Score:** 4

**Q1 Summary And Contributions:**

The paper considers a case of objects represented by unordered sets of tensors (e.g., point clouds, or graphs). Standard DL architectures use an aggregation layers to convert an input representation into a fixed-dimensional vector. The paper proposed an aggregating function that is a solution of an optimization problem. The authors proved a universal approximation theorem. Also, they demonstrated that such aggregation function can be useful when used in GNNs.

**Q2 Assessment Of The Paper:**

More detailed information regarding each of these aspects is given below:

**Q2(4) Quality Of Experiments (Optional):**

2: Fair: The experimental evaluation is weak: important baselines are missing, or the results do not adequately support the main claims.

**Q2(5) Reproducibility:**

2: Fair: Key resources (e.g., proofs, code, data) are unavailable but key details (e.g., proof sketches, experimental setup) are sufficiently well-described for an expert to confidently reproduce the main results.

**Q3 Main Strengths:**

- the paper proposed some further generalisation of an aggregating layer
- the authors proved that the proposed equilibrium aggregation can approximate permutation invariant continuous functions
- some experiments with GNNs demonstrated better performance when using the proposed aggregation layer

**Q4 Main Weakness:**

- the main weakness is an experimental evaluation. The authors considered only 1) one toy problem 2) one test problem 3) one dataset and some standard GNNs with the proposed aggregation approach

**Q5 Detailed Comments To The Authors:**

The proposed generalisation of an aggregating function is novel and has a universal approximation property, namely, it can approximate permutation invariant continuous functions.

However, the experimental evaluation of this building block is very limited.

- the authors did not make any ablation study on how the complexity of the function F, used in the aggregation layer and modelled by a neural network, influence the efficiency of the aggregation and the accuracy of the final model
- the authors did not make any investigation on how the optimization algorithm and its properties (e.g. the number of iterations), used to solve the optimization problem in eq. (2), influence the final results
- the authors considered only one (!) complex real-world dataset containing graphs to test the proposed layers inside two standard GNNs

I think that
- more datasets and architectures should be considered to demonstrate usefulness of the proposed approach
- various ablation studies should be done
- other types of data should be considered, e.g., point clouds

The authors did not provide a detailed study about to what extent the proposed modification makes the learning process slower.

In (5) the authors mentioned some auxiliary loss. How does it influence the results? Results with the loss and without it?

The authors mentioned some source code in the Appendix, but there is no source code to reproduce the results of the experiments.

The authors did not compare the proposed approach with any baseline approaches, mentioned in the related section. E.g. what is about a comparison with the results of the Deep Sets approach?

page 6, "We compare three models, Sum aggregation inspired by [?]," - reference is missing.


**Q7 Justification For Your Score:**

The idea is good. Unfortunately, the experimental evaluation is very weak. This is the main reason to reject the paper.

=============

I carefully read other reviews and the author's response. It is good that the authors provided some ablation on how the hyperparameters influence the final results. So I decided to increase the grade slightly. However, to convince the practical usefulness of the method, applied results are needed. I insist that one real-world benchmark is not enough.

**Q9 Complying With Reviewing Instructions:**

1: Yes.

---

### Official Review · Reviewer_VmK1 · 2022-04-26

**Q2(1) Originality/Novelty:** 3
**Q2(2) Significance/Impact:** 2
**Q2(3) Correctness/Technical Quality:** 3
**Q2(6) Clarity Of Writing:** 3
**Q6 Overall Score:** 5
**Q8 Confidence In Your Score:** 4

**Q1 Summary And Contributions:**

The paper propose a method to learn aggregation functions for networks that combine sets of information (a set as in Deep Sets, or the global aggregation phase of a graph network), specifically by learning an energy function to be optimized with gradient descent. Universality of the scheme is demonstrated, and it is shown to work well in a few experimental settings.

**Q2 Assessment Of The Paper:**

More detailed information regarding each of these aspects is given below:

**Q2(4) Quality Of Experiments (Optional):**

3: Good: The experimental evaluation is adequate, and the results convincingly support the main claims.

**Q2(5) Reproducibility:**

3: Good: Key resources (e.g., proofs, code, data) are available and key details (e.g., proofs, experimental setup) are sufficiently well-described for competent researchers to confidently reproduce the main results.

**Q3 Main Strengths:**

- Sensible but novel idea

- Some theoretical justification (universality of the results)

- Good empirical results

**Q4 Main Weakness:**

- There is one obvious algorithmic change that could potentially be a significant improvement, but isn't considered
- The theoretical results are quite limited (see detailed comments)

**Q5 Detailed Comments To The Authors:**

## Architectural change

Why did you not consider input-convex neural networks (Amos et al. 2017, already cited)? Given that you clearly know about the work (it's the title of the paper you cited...), this seems like a very strange omission: it would avoid most of the concerns about the ability to optimize the energy, and may also help with the concern about learning rates mentioned in Appendix C.5, since the learning rate should be less important. (Implicit differentiation may still be less effective than explicit unrolling, since its accuracy generally requires reaching a fairly accurate solution, though.)


## Theoretical limitations

The approximation error result is, I think, fairly unsurprising and somewhat limited. It's a good property to have, but it is a purely representational result, and not even an explicit one: it simply shows that there is a particular energy function that works: one which, I think, cannot be exactly realized by a ReLU network, but can be uniformly approximated on a compact domain by a wide (and/or deep) ReLU net.

No results on the estimation or optimization error are provided. A uniform convergence-type result would likely be achievable here, and might end up quite informative – I'd recommend trying to prove such a bound if you haven't. An optimization result would likely be pretty difficult, so it's not surprising nothing is provided there.

The known theoretical advantage of Equilibrium Aggregation over Deep Sets is also very limited (section 3.2). In particular, the focus here is comparing when the latent size is very small – but this ignores that Equilibrium Aggregation has a significant new set of parameters used in a complex way to be learned: is it really fair to compare the two? Can the max / mean energy functions actually be learned with small ReLU nets? (Looking at Table 1, the max function $F(x, y) = \max(0, x - y)$ can be represented very compactly with a ReLU net, but the mean function $(x - y)^2$ maybe can't.)


## Miscellaneous minor points

- The paper of Rajeswaran et al. is duplicated in your bibliography.

**Q7 Justification For Your Score:**

I think the core idea of the paper is sensible and to my knowledge fairly novel in the area, with potential for impact on set- and maybe graph-based architectures. But I'm not convinced this is the best argument for the idea, and I'd really like to see results with input-convex networks for the energy function.

**Q9 Complying With Reviewing Instructions:**

1: Yes.

---

### Decision · Program_Chairs · 2022-05-15

**Decision:**

Accept (Poster)

**Comment:**

Meta Review: The focus of the submission is encoding sets, particularly with neural networks. The authors propose to use a potential function (F_\theta) based generalization of standard aggregation schemes [as formulated in (2)]---referred to as equilibrium aggregation---where F_\theta is a neural network with parameters \theta. They show (Theorem 1) that this aggregation scheme can be universal for some choice of F_\theta when applied on scalar inputs belonging to [0,1]. The efficiency of the approach is illustrated in median estimation, omniglot class counting and molecular property prediction.

Learning on sets is an important area of machine learning, with various applications. The authors present an extended tool (a generalized aggregation scheme) in this context. Though the theoretical understanding of the proposed equilibrium aggregation is limited, the paper can have some practical impact.

The submission would benefit from putting it into the wider context of machine learning on probability measures (distribution classification, distribution regression, ...).